# Sizing Milking Groups in Small Cow Dairies of Mediterranean Countries

**DOI:** 10.3390/ani10050795

**Published:** 2020-05-04

**Authors:** Angela Calvo, Gianfranco Airoldi

**Affiliations:** Department of Agricultural, Forest and Food Sciences (DISAFA), University of Torino, Largo P. Braccini 2, 10095 Grugliasco, Turin, Italy; gianfranco.airoldi@unito.it

**Keywords:** milking system, work routine, parlor, milking model, small dairy

## Abstract

**Simple Summary:**

The dimensioning of cow milking systems has been studied for many years by many authors, but nobody has addressed the studies on small cow dairies that are actually present in many Mediterranean countries (with a number milked cows up to 100–120). The number of cows, the financial resource, the skilled workforce and the availability of time are parameters influencing the selection of the milking system also in smaller herds. What is actually lacking in these dairy farms is milking unit dimensioning, whatever the used milking system. This study analyzed the milk routines performed by milkers with different milking systems in small dairies (tie-stall and little parlors). The work-flow analysis was the starting point to develop different models for predicting the optimal milking unit sizing.

**Abstract:**

A dairy farmer chooses the number of milking groups in function of the herd size, stall type and milking system also in small cow dairies (number of animals lower than 100–120). In these dairies, there are different milking systems (bucket, trolley, pipeline, little autotandem, herringbone or parallel parlors) and each of them has a different work routine. The knowledge of the routine is the starting point for assessing the milking installation, because it determines the number of milked cows per hour. Different milking systems have common tasks (as pre-dipping, inspecting foremilk, udder preparation, attaching teat cups, post-dipping), but in the meantime there are different operations that characterize each specific routine (e.g., animal entry and exit if there is a parlor, bucket, trolley or milking group positioning if tie-stall). For this reason, we surveyed twenty small dairy farms located in the Piedmont Region (Italy) with different milking systems to correctly acquire the specific milking routines. Different models were therefore studied using the observed routines in in the examined farms. These models were then used to calculate the number of milked cows per hour and the number of milking groups. The main findings were simple equations, specific for each milking system, easily accessible by the farmer to correctly size his milking system.

## 1. Introduction

Mechanical milking consists of a logical sequence of straightforward repetitive tasks for the milkers (after preparing the milking equipment and, when necessary, the parlor): animal approaching, udder preparation, pre-dipping, attaching the milking unit, waiting for the milk flow, detaching the milking unit (if the automatic detacher is not present), post-dipping [1].

The main concern of the mechanical milking was at the beginning to increase the efficiency of the milking process to improve milk quality and quantity, and the labor efficiency [2,3,4,5]. Some studies introduced the time and motion method to establish the work routine [3,6] because this methodology provides a remarkable measurement of performance, and it is also the baseline for the milking mathematical models [7,8,9]. Computer-assisted simulation models of milking parlor performance were therefore accomplished since the seventies to improve the milking throughput and the labor performance [10,11,12], especially to reduce the daily routine labor and consequently the related milking costs [13,14]. In the following years, the researchers addressed their studies on mechanized milking systems also to the operators’ and animals’ welfare, considering the effects on the routine of the milking tasks [15,16,17]. Some studies, for example, found that the optimum time to attach teat cups had benefits on the teat condition as well as on a higher milk yield [18,19]. The main goal of more recent studies was also to forecast the optimum parlor [1,20], and to assist the farmer for the most efficient parlor design [21] and management [22,23,24], because the milking center is the most expensive item for a dairy farm [25].

In compliance with the herd size and with the stall type (free or tie), the farmer must select the milking system [17], and consequently the number of the milking units and the necessary labor force. The main aspect to work out for a rational decision is the work routine of the milking operation. The time occurred for both the pre-milking and post milking tasks is essential to determine the number of milking stalls and milking units that a milker can manage in the operation timeframe. Milking routine defines how one or more milkers perform a mechanic milking procedure over multiple cows (independently by the animal stalling). The preliminary knowledge of the usage of time for preparing and finishing the milking determines the number of cows that can be milked per hour and therefore the precondition for assessing milking installations [26].

Many authors assessed the need to increase the efficiency of the milking process through the optimization of milking routine and some of them applied their results mainly to the parlors (parallel, herringbone and rotary) [10,11,14,20,23].

For many decades, mechanical milking is spread everywhere, in a different type of dairy farms (cows, sheep, goats, buffaloes and others). It is nowadays common also in traditional small-scale dairy farms of the Mediterranean areas [27,28,29], where different (and sometimes outdated) milking systems are present (bucket, trolley, tethered, small parlors). As in large herd sizes, though also in smaller herds, the number of cows, the financial resource, the skilled workforce and the availability of time are parameters influencing the selection of the milking system. What is actually lacking in these dairy farms (with a number milked cows up to 100–120) is an indication for the sizing of the necessary milking groups, whatever is the used milking system. The main goal of this work was therefore to suggest different models for predicting the optimal milking unit dimensioning to address the farmers to the best choice. The modeling is an alternative of the stiff tables of manufacturers, which are not always flexible in the adaptation to specific small dairy needs.

## 2. Materials and Methods

### 2.1. Preliminary Assumptions

Achieving good and realistic results in milking operations in small cow dairies, both in labor productivity and animal welfare, only one experienced and qualified milker, familiar with the physiology of dairy cows, was used. The milking systems were moreover all timely revised and accurately washed after each milking.

### 2.2. Milking Extraction Time (Milk Flow Time)

Many studies reported the effect of milk flow time on milking performance [4,30,31] as the most influencing activity. Even though the milk extraction is not a real operation of the routine, its duration highly influences the work routine time. For example, the milking time could be the key element of the operator’s idling time in case of cows with high milk flow time, or it may cause a dangerous over milking in the opposite situation, when a cow has a lower milk flow time than other animals and automatic cluster removals are not present.

The milking extraction time of each cow depends on multiple variables: animal milk yield, correct feeding and stimulation during the preparation tasks, the physiologic predisposition of the cow to release milk more or less quickly.

Since the seventies, many authors studied and simulated the milking extraction time, and they found average times around 5–5.5 min [2,6,10,32,33]. In the following decades, however, the milk yield per milked cow increased, thanks to the genetic improvement of the animals, and thereby the milking extraction time increased [30]. New relationships between the total milking time and the milk yield were assessed. As things stand, a milking extraction time between 6 and 8 min is reasonable [34].

### 2.3. Milking Routine: Definitions

The milking routine is the amount of the operator’s tasks performed during the milking of each cow. Each task requires a certain amount of time, and the sum of these times in addition to the milking extraction time determines the number of cows that the operator can milk per hour. The work routine is defined as the function of the amount and type of the milking equipment, the operators’ skills and the duration of the milking operations [3].

Even though each milking system has its work routine with specific sequences of operations, there are some common tasks, mostly manually performed by the milker: pre-dipping, inspecting foremilk, udder preparation, attaching teat cups, removing teat cups (if the milking cluster removal is absent), post-dipping (disinfecting teats). Some definitions are necessary before introducing the proposed models for sizing the milking system in the function of the tie-stall or parlor type (Table 1).

The milking routine *Tr* (the time used by the operator for milking each cow) is split into the following categories: early (initial) routine (*Tri*, Equation (1)), ending (final) routine (*Trf*, Equation (2)) and downtime (*Td*), always present whatever the milking system. The entry (*Ten*) and the exit (T*ex*) of the animals, in and out of the parlor, are common parameters in *Tri* and *Trf* in parlors. Bucket, trolley, or cluster positioning (*Tpos*) is a *Tri* parameter in tie-stall, as the emptying of the bucket or tank in the trolley (*Temp*t) in *Trf*. The relocation of the mobile parts of the milking system (*Trr*) must be instead included in the final routine *Trf* in the trolley system (Equation (2)).
*Tri* = *Ten** + *Tpos*** + *Tpre* + *Tfor* + *Tup* + *Tatt* (min)(1)

* only parlor, ** only tie-stall.
*Trf* = *Trem* + *Tpost* + *Tempt** + *Trr*** + *Tex**** (min)(2)

* only bucket and trolley, ** only trolley, *** only parlor.

Downtimes are always present in each phase of the milking routine and must be considered in the calculation of the total routine time, *Tr* (Equation (3)).
*Tr* = *Tri* + *Trf* + *Td* (min)(3)

The output milker’s time, *Tc,* for milking each cow depends on the milking system and includes the calculated routine time, *Tr* (Equation (3)), the milk flow time, *Tm*, and, when present, the possible unproductive times, *Tmw* (as the milker’s waiting time when the automatic cluster removal is absent). Equations (1)–(3) are used whatever the milking system. The washing and the maintenance of the milking unit, as well as the transportation of the milk tanks outside the stall at the end of the milking operation in tie-stall, were not considered in the calculation of the routine time, *Tr*, since it is strictly related to the milking tasks with the cows.

### 2.4. Inspected Milking Systems: Tethering Cows

In tie-stalls, cows are constrained to their stall barns, where they are directly milked. Tethering cows are still present in little dairy farms of many Mediterranean countries [35], where the milking is often carried out with trolley or bucket milking systems. In both these systems, the milker’s time for each cow is greater than the routine time (*Tc* > *Tr*), because he/she has to wait for the completion of the milk flow time before moving to the next animal. If a milk-line (stall barn with pipeline) is not present, the labor productivity is low, as the milker (besides the basic routine) must move the trolley or the bucket alongside the next cow and transport the bucket outside the barn when full.

### 2.5. Inspected Milking Systems: Little Parlors

There are many types of parlor, the choice of which depends on the herd size and characteristics, the economic impact on the dairy, the number and the ability of the workers, and the automation level [2,5,25]. Parlor type affects building size, cow traffic to and from parlor, milking routine and mechanization level. The number of the daily milking is also important, for example, in the small cow herds of Italy, two milking per days are planned and animals are milked every about 12 h (usually at 4:00 and at 16:00). Two milkings/day represents a standard situation in small cow herds, not only in Italy. As also observed by other authors [36,37], it is convenient to maintain the same milking intervals during the day (every 12 h) in the cows, because milking time is one of the components that modulates the milk lipolytic system.

Parlors may be individual and collective (batch milking). In the first type, cows stand nose-to-tail inside individual stalls (tandem and autotandem parlors). In the second, cows are milked together after their entry in the milking stalls. In collective parlors, the milking operations are sequentially carried out, and it is important that the milking extraction time is almost the same for each animal. The swing-over milking system permits to shift the milking unit from one stall to another positioned in front of it, speeding up the milking routine (Figure 1).

A brief description of the milking systems is provided to understand the different modelled routines in the result chapter. Rotary parlors, which are more complex and overly expensive in small dairy herds of Mediterranean countries, were not considered in this work.

#### 2.5.1. Tandem (Side-Opening) and Autotandem

In this type of parlor, the cows move individually to their milking stalls entering and exiting through gates manually (tandem) or automatically (autotandem) opened up and closed (Figure 2). The milking operation is performed individually, and cows may have a different milk flow time without affecting the milking routine of the other animals.

#### 2.5.2. Herringbone

In herringbone and parallel parlors, cows are handled in groups. The size is variable from 4 × 2 to 30 × 2 [25]. Many herringbone and parallel parlors are equipped with rapid-exit stalls (by freeing all cows at once from one side of the parlor in a direction perpendicular to the entry lane, Figure 3), which increase the milking parlor efficiency and improve the milking routine.

#### 2.5.3. Parallel

Parallel parlor resembles the herringbone, but cows are arranged perpendicular to the edge of the milker’s pit, where he/she works back to the animal (Figure 4). Advantages of this system for the lowering of the milking routine are: higher displacement of the cows (animals move faster) and fewer movements of the milker. As for herringbone, also in this case, a milking group for each stall or a milking group every two stalls (swing-over) may be provided.

### 2.6. Study of the Models

The studied models were based on the available literature [8,11,20,25,37,38,39], following the indication provided by Armstrong et al. [3], and analyzing the work flow during the milking operation in twenty small cow dairy farms located in north-west Italy. The dairy farms were located in the Stura mountain valley, in the Monferrato hills and in the Po valley in Piedmont Region. Piedmontese cows were present in the farms with trolley and bucket systems, while the Italian Friesian breed were in the other farms equipped with the milk-line and the parlors. The distribution of the surveyed milking systems was: three buckets, three trolleys and four milk-lines in tie-stalls, two autotandem, three herringbone and five parallel in parlors. The milking operations in each farm were surveyed for three days, while the number of lactating cows and milk yield per day per cow were furnished by the farmer.

### 2.7. Data Elaboration

Each routine task (pre-dipping, post-dipping, foremilk inspection, teat cups attach and so on) has a short duration, and, for this reason, they were acquired and studied using seconds. Initial and finale routines *Tri* and *Trf* were instead counted in minutes (with seconds expressed in hundreds), to have the same dimension unit of the milk extraction time (as used by other Authors) and to make comparisons with other researches. Collected data in the twenty dairy farms were analyzed using IBM SPSS Statistics (version 25, International Business Machines Corporation, Armonk, New York, NY, USA). The GLM (general linear model, to analyze quantitative data and to understand how the mean response relates to one or more independent predictors) was used to assess the effects of the variable milking system on *Tri* and *Trf* routines, with a confidence level *p* = 0.01. The homoscedastic condition (assumption of equal variance) was previously tested by the Levene’s test.

## 3. Results

### 3.1. Tie-Stalls

#### 3.1.1. Trolley

In trolley systems, all components (vacuum and milk system) are positioned on a mobile frame and transported inside the barn (Figure 5). One or two milking groups (maximum) are housed on the trolley, managed by one operator.

The time *Tc* used by the milker for a single cow is calculated as in Equation (4), while the number of milked cows per hour is obtained by Equation (5).
(4)Tc=Tr+Tm
(5)$$Nc=\frac{60}{Tc}$$

With two milking units, the milking operation can be simultaneously performed by the same operator on two cows, but he/she must wait the ending of the milk flow regardless. In this case, an unproductive time is present (*Tmw* = *Tm* − *Tri*), to the detriment of the milking routine.

#### 3.1.2. Bucket

In bucket systems, the motor unit and the vacuum system are located in a special room near the stable. The mobile milking buckets are connected to the vacuum pipeline (Figure 6), and the operator must transfer the milk from the buckets to the transport bulk tank and move it out of the stable. He/she is able to control up to three milking groups.

The number of milked cows per hour *Nc* depends on the number of the milking groups *Ng* managed by the operator (Equation (6)). In this equation, *Tm/Tr* is usually a decimal number that must be rounded up or down. The rounding up (*ru*) may cause the serious problem of over-milking the cow, and, for this reason, in this milking system it is better to round down (rd) the fraction *Tm/Tr*.
(6)$$Ng=rd\left(\frac{Tm}{Tr}\right)+1$$

Equation (7) gives the time *Tc* dedicated by the milker to each cow, while the number *Nc* of milked cows per hour per group is given by Equation (5).
(7)$$Tc=Tr+\frac{Tm-Tr\times (Ng-1)}{Ng}$$

#### 3.1.3. Milk-Line

The milk-line has two rooms near the barn: a machine room and a milk room. There are two fixed pipes for milk and for vacuum (Figure 7).

The number of milking groups *Ng* and the milker’s time per cow *Tc* are once again calculated by Equations (6) and (7) if the milking cluster removal is not present. With the milking cluster removal, the number of groups is calculated using Equation (8), rounding up the ratio *Tm/Tr*. The cow may wait for the final routine without any over-milking risk (Figure 8).
(8)$$Ng=ru\left(\frac{Tm}{Tr}\right)+1$$

If the milking cluster removal is not present, the number of milked per cows per hour *Nc* is calculated by means of Equation (5), otherwise (with automatic milking removal) with Equation (9).
(9)$$Nc=\frac{60}{Tr}$$

Table 2 gives the routine for bucket and trolley systems, while Table 3 shows the same routine for the milk-line without and with the automatic cluster removal (NOACR and YESACR respectively). The routine times were obtained by directly observing the milking operations in the examined diaries and then comparing them with the bibliography [38,39,40,41,42].

The initial routine *Tri* in the bucket system is close to the *Tri* in the milk-line without automatic cluster removal, while the final routine *Trf* is definitely shorter in all the milk-line systems (here, there are not buckets to empty). The automatic cluster removal, other than eliminating the over-milking risk, reduces the initial routine *Tri* by 20%, and the final routine *Trf* by 50% (Table 4).

### 3.2. Parlor

In the parlor, the milking time per cow, *Tc,* is always equal to the routine time, *Tr*, while the number of milking groups (and, consequently, the number of milked cows) depends on the milk flow time *Tm* and on the routine time *Tr*. The milking cluster removal is supposed to always be present in the examined dairies to avoid over-milking, as it is necessary to work with many animals simultaneously. The initial routine, *Tri,* and the final routine, *Trf,* are shorter than in tie-stalls, as there is no need to move trolleys, buckets or milking groups. After having performed the initial routine on the first cow, the milker can move on to the second animal for the initial routine and so on, until the first cow has finished being milked. The milker then returns to the first cow and carries out the final routine.

#### 3.2.1. Autotandem (Side-Open) with Automatic Milking Cluster Removal

In side-open parlors, the swing-over is not present, because animals are handled individually and not in groups. The number of milking units, *Ng,* is given by Equation (8), because the automatic milking removal is present. If *Ng* is an odd number, the shrewdness is to add a further milking unit, because the number of stalls is even (cows are positioned on two parallel rows). *Tc* is equal to *Tr* and *Nc* is given by Equation (8). Figure 9 portrays the routine in the examined autotandem parlor (six stalls and *Te* = 6 min).

#### 3.2.2. Herringbone and Parallel Parlors

Initial and final routines are the same as in autotandem, but the entrance of the animals and the pre-dipping operations take less time, as the operations are sequentially performed (collective animal milking) on two parallel rows. Without swing-over, the number of milking groups *Ng* (equal to the number of stalls *Ns*) is calculated distinguishing if *Tm/Tr* is an integer number or not (Equation (10)).
(10)$$\begin{array}{l} \frac{Tm}{Tr}=INT\left(\frac{Tm}{Tr}\right)\to Ng=\frac{Tm}{Tr}\times 2\\ \frac{Tm}{Tr}\neq INT\left(\frac{Tm}{Tr}\right)\to Ng=\left(INT\left(\frac{Tm}{Tr}\right)+1\right)\times 2 \end{array}$$

When the swing-over is present, the number of milking units *Ng* (half of the number of stalls *Ns*) is now half of the previous calculated by Equation (10), while Equation (11) gives the number of stalls *Ns*. The number of milked cows per hour *Nc* is in Equation (9).
(11)Ns=2×Ng

The milking routine in parallel parlors is the same as herringbone, but with a lower entry time of the animals (and consequently with a lower *Tr*). The milking routine observed in a parallel parlor with 16 stalls and *Te* = 8 min is depicted in Figure 10.

Table 5 shows the initial and final routines in the examined parlors: autotandem, herringbone and parallel.

Final routines, *Trf,* are almost the same in all these parlors, while there is a difference between the individual (autotandem) and collective parlors in the initial routine, *Tri* (Table 6). In herringbone and parallel parlors, in fact, animal entry and pre-dipping are fastened by the group management of the cows.

### 3.3. The Examined Dairy Farms

We examined the milking tasks performed in different dairy farms with the same milking system (three trolleys, three buckets, four milk-lines, two autotandem, three herringbone and five parallel), and the most interesting result of the survey was that the milker executed the same initial and final routines in the farms with the same milking system.

#### 3.3.1. Tie-Stall Dairy Farms

The number of lactation cows was between 22 and 45 in the farms with trolley, bucket and milk-line systems without automatic cluster removal, increasing to 50 animals in the farm with the milk-line equipped with the automatic cluster removal (the sole investigated, because this milking system is not spread in the Piedmont Region). The milk flow time was in the range 5.5–7.6 min, with higher values in the milk-line systems (Table 7).

Concerning the daily milk yield, some differences were observed (Figure 11).

In the farms with trolley and bucket systems, the milk yield was around 18.7 L per day, against an average of 32.5 L per day in the farms with the milk-line system. This difference is due to the breed present in these last farms (Italian Fresian cows), more productive than the Piedmontese breed cows [43].

#### 3.3.2. Dairy Farms with Parlors

From 95 to 130 lactating cows (Italian Fresian breed cows) were present in the dairy farms equipped with the parlor. All these farms, but three (located in the Stura valley), were located in the Monferrato hills and in the Po valley near Cuneo.

The observed milk flow time was between 5.9 and 8.1 min, with lower and more homogeneous values observed in herringbone and parallel parlors (Table 8).

The milk yield per cow (l day^−1^) was always around 40 L per day, with peaks of 50 L in two farms with the parallel parlor and the highest number of lactating cows (more than 120). A slightly lower production (around 32 L) was observed in two farms with the autotandem milking system and with the lowest number of lactating cows (less than 100).

#### 3.3.3. Differences in the Observed Routines

Milking systems statistically influenced both the initial (*Tri*) and final (*Trf*) routines (Table 9), due to the high differences between tie-stall and parlors routines (Table 4 and Table 5).

On the other hand, the Tukey post-hoc test highlighted similarities among some milking systems (Table 10) for both *Tri* and *Trf*. The highest statistical significances were found in *Tri* of parallel and herringbone parlors (0.70 and 0.75 average minutes, respectively), and in trolley and milk-lines with automatic cluster removal (1.26 and 1.30 average minutes). *Trf* were significantly the same in parallel (average 0.23 min), herringbone (average 0.25 min) and autotandem parlors (average 0.27 min). Bucket and trolley milking systems had similar *Trf* times (1.57 and 1.68 min), but with a lower statistical significance (0.61).

## 4. Discussion

The efficiency of a milking system is usually assessed on the basis of multiple parameters, which, in our situation (small herds with a number of animals not higher than 120), are reduced in cows milked per hour and the number of stalls to complete the milking in time (about 2 h). For this reason, the focus was to discuss the above-mentioned results, as obtained by the models and by the surveyed farms.

### 4.1. Tie-Stall

In Table 11, there are the calculated parameters *Tr*, *Tc*, *Ng* and *Nc* in two different scenarios: milk flow time *Tm* = 6 min and *Tm* = 8 min.

The trolley system is less convenient than the bucket system because it takes more time and the number of milked cows per hour is lower (due to the fact that the milker has to wait the complete the milking of two cows before moving the whole trolley to the following animals). In both cases, from the routine point of view, the milker has many unproductive times (*Tmw*), Figure 12.

The number of the milked cows per hour was 13.46 for the trolley system and 26.23 for the bucket (Table 11), allowing the milking of the lactating cows in the surveyed farms, 24.7 and 34.3 in average, also considering the high standard deviation (Table 7).

The milk-line provides, as expected, a more efficient milking, especially when the automatic cluster removal is present. In this case, the number *Nc* of milked cows per hour (34.29) does not change, whatever is the milk flow time (Table 11), because it is calculated using only the routine time *Tr*, independently from the milk flow time. Table 11 also shows a firm decrement of the time dedicated by the milker to each cow in the milk-line when the automatic cluster removal is present (-44.6% if *Tm* = 6 min, −37.5% if *Tm* = 8 min), as also found by Rasmussen [41]. The presence of the automatic cluster removal also significantly decreases the over-milking caused by a prolonged teat stimulation, with the risk of cause teat congestion [44]. The number of calculated milked cows per hour (Table 11) with the milk-lines were 21.43 (NOACR) and 34.29 (YESACR), which allowed for the milking of the observed 35 and 50 mean cows (respectively) in time.

### 4.2. Herringbone and Parallel Parlors

In this example, the routine time *Tr* is 1.27 min in autotandem, 1 min in herringbone and 0.92 min in parallel (Table 12). The preparation time (initial routine *Tri*) in herringbone and parallel parlors is similar to the values recommended by Peychev et al. [38] (between 45 and 90 s, Table 5). Without considering the animal entry, the *Tri* calculated in herringbone and parallel parlors (Table 5) are in line with the data found by Smith et al. [39] (32 s).

As *Nc* is directly obtained by *Tr*, regardless of the milk flow time, the number of milked cows does not differ in the two scenarios with a different milk flow time (6 and 8 min). The autotandem is less efficient than herringbone and parallel parlors, since the time of both animal entry and pre-dipping is greater (cows move individually and they are not milked together).

Higher is the number, *Nc,* of milked cows per hour, and lower is the routine time, *Tr*; this is a self-evident consequence of the calculation of *Nc*, but the numbers are confirmed also by other authors, independently by this equation [4,45]. The number of milked cows per hour in the double-8 herringbone (60) is slightly lower than 64, as found by Burks et al. in 1998 [1], but is in line with Krumm et al. [23], and the 65 cows per hour of double-9 parallel are compliant with Hatem et al. [20].

Carreira et al. [42] studied and developed some algorithms for the best sizing of herringbone and parallel parlors, starting from the routine time and the milk flow time, as we did in this work, but without distinguishing the parlor type. As also observed here, they got similar results (Figure 13), recording any change when the number of clusters per milker was greater than (or equal to) 12 (we indeed observed two different values in the number of milked cows per hour with 14 and 18 clusters, due to the slightly differences in the routines of herringbone and parallel parlors, assumed to be equal by the other authors).

The milker expertise and the type of milking facility affect labor and routine efficiencies: for this reason, the input values used in these models may change; nevertheless, the equations remain valid. In the farms equipped with parlors, the number of calculated milked cows per hour *Nc* (Table 12) allowed for the milking of all the lactating cows present in the surveyed farms (Table 8) in time.

## 5. Conclusions

Even though it is not easy to find literature concerning the milking systems in tie-stall and little parlors, there are still many dairy farms in Mediterranean countries with a low number of cows (less than 100–120) that cannot face high investment costs in expensive milking plants. It would appear apparently naïve modelling milking with low numbers of animals, but there is still a lack in the work organization of these small herds, with few people working a great amount of hours per day. For this reason, we surveyed twenty small dairy farms located in Piedmont Region (Italy) equipped with different milking systems to correctly acquire the milking routines. Different models were therefore studied using the observed routines in the examined farms. These models were developed independently by the routine times measured in each farm. The measured times were the method to validate the number of milked cows per hour and the number of milking groups in these small dairies. The simple models presented in this paper could therefore support the breeder to correctly dimension and choose the most effective milking system for his dairy farm.

## Figures and Tables

**Figure 1 animals-10-00795-f001:**
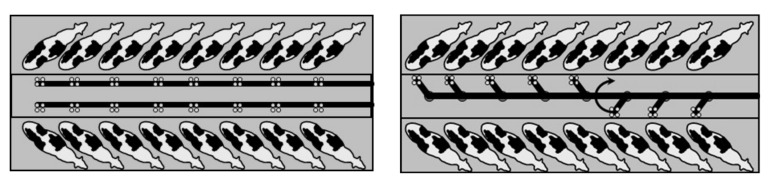
Herringbone parlor without (left) and with milking swing-over milking system (right).

**Figure 2 animals-10-00795-f002:**
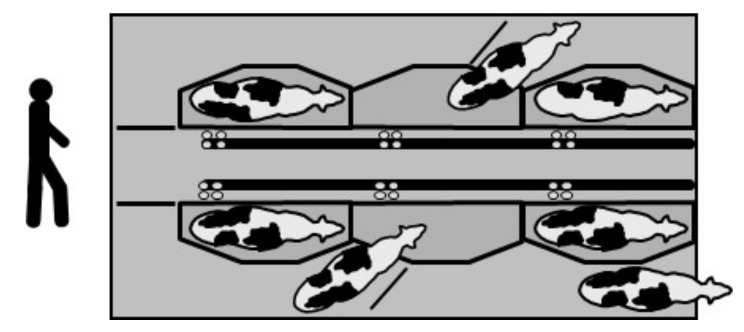
Autotandem parlor with 3 + 3 stalls.

**Figure 3 animals-10-00795-f003:**
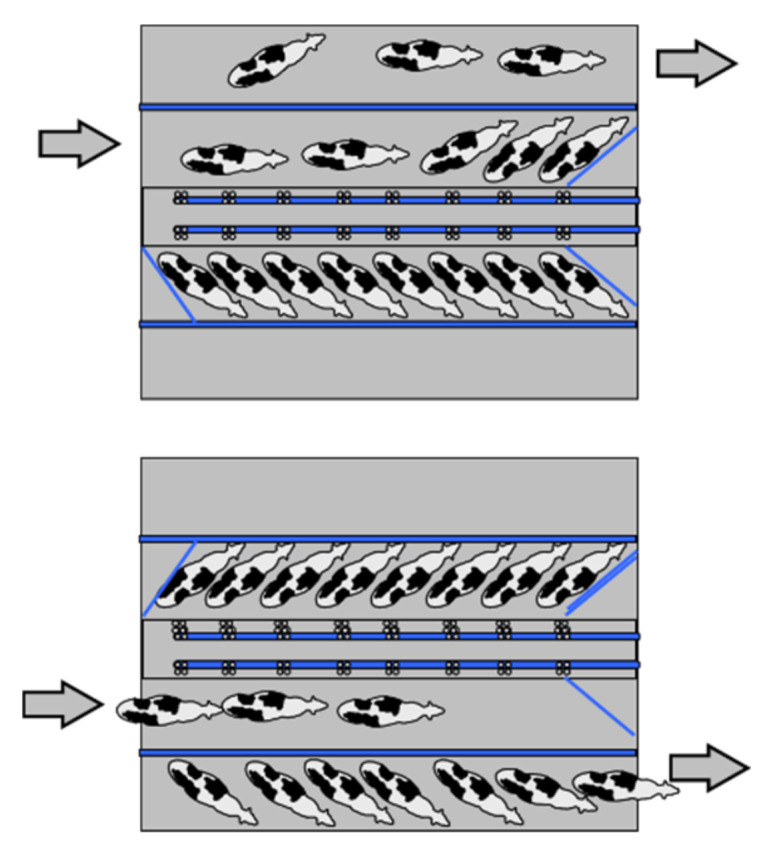
Herringbone parlor with rapid-exit system.

**Figure 4 animals-10-00795-f004:**
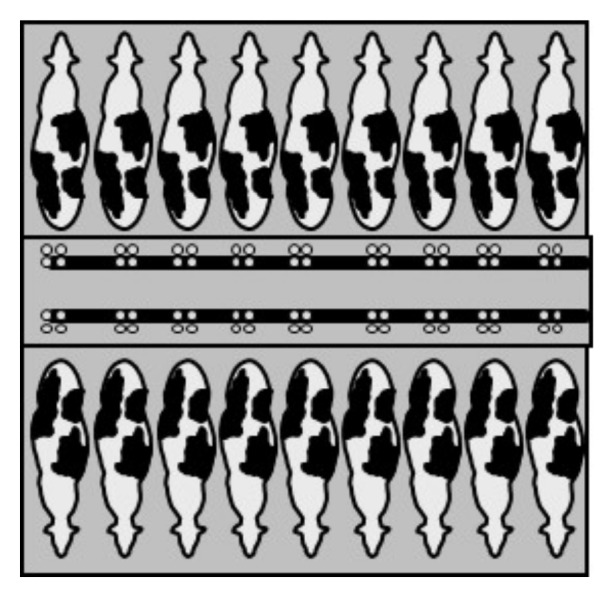
Parallel parlor.

**Figure 5 animals-10-00795-f005:**
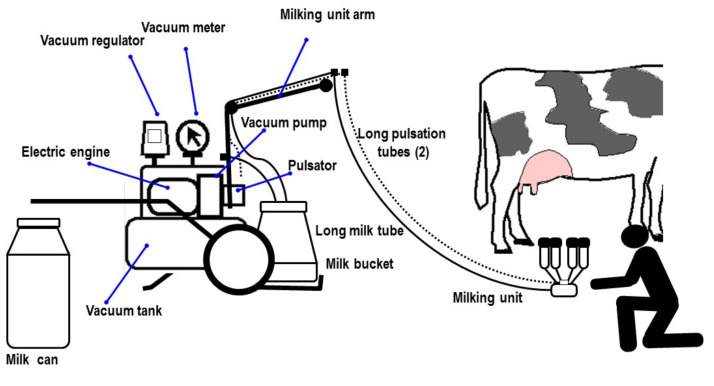
Trolley milking system.

**Figure 6 animals-10-00795-f006:**
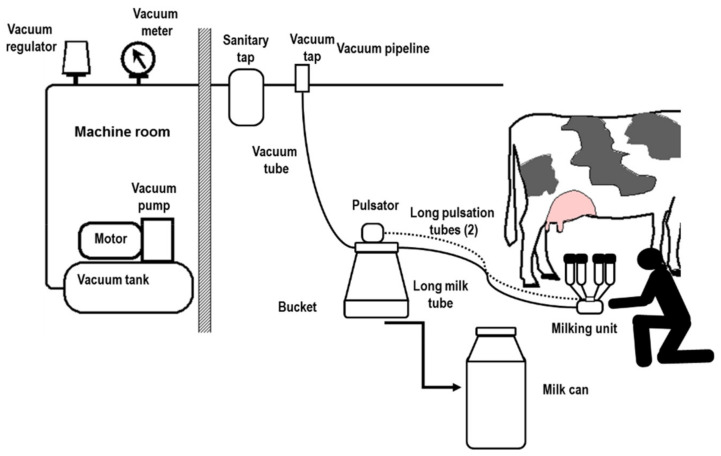
Bucket milking system.

**Figure 7 animals-10-00795-f007:**
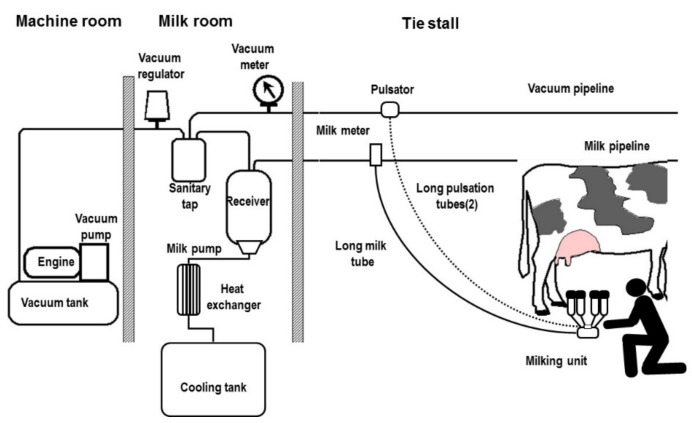
Milk-line.

**Figure 8 animals-10-00795-f008:**
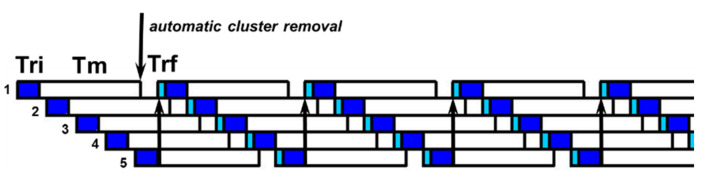
Milk-line routine in the presence of milking cluster removal (Ng = 5 and Tm = 6 min).

**Figure 9 animals-10-00795-f009:**
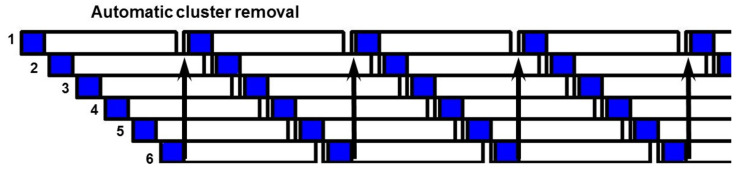
Example of a milking routine in an autotandem parlor (3 + 3 stalls and *Te* = 6 min).

**Figure 10 animals-10-00795-f010:**
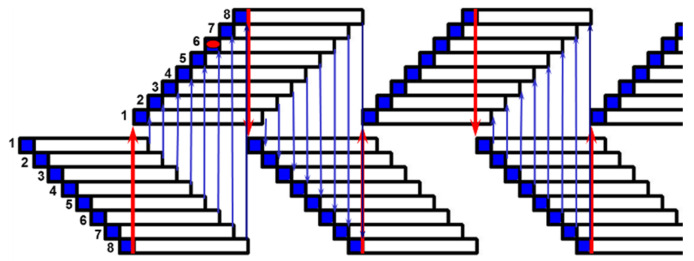
Milking routine observed in a parallel parlor (8+8 stalls and *Te* = 8 min).

**Figure 11 animals-10-00795-f011:**
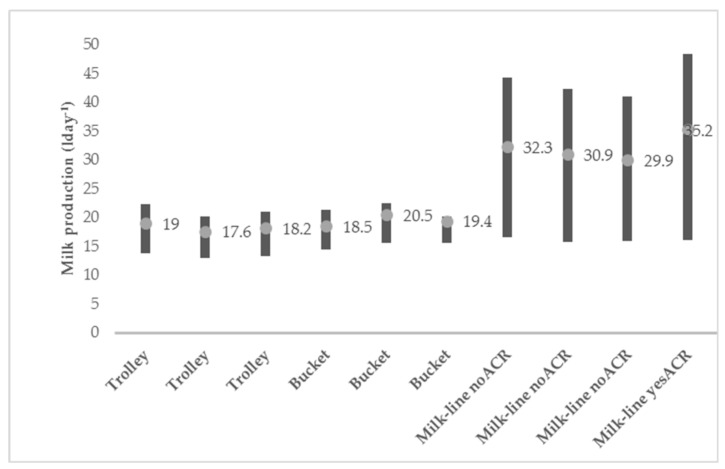
Milk production (min, max and mean) in the surveyed tie-stall dairy farms (lday^−1^).

**Figure 12 animals-10-00795-f012:**
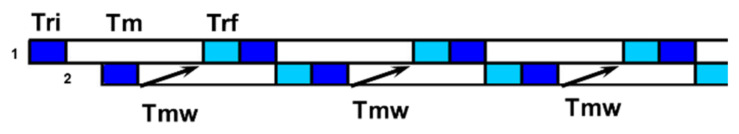
Example of milker’s waiting time, *Tmw,* when using the bucket milking system.

**Figure 13 animals-10-00795-f013:**
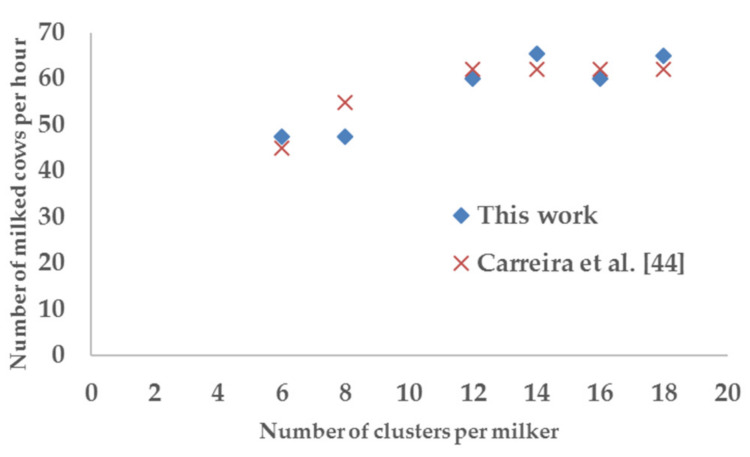
Number of milked cows per hour in function of the number of clusters per milker (comparison between this work and the study of Carreira et al. [42]).

**Table 1 animals-10-00795-t001:** Parameters used in the milking sizing model.

Simbol	Unit	Meaning	Further Information
*Nc*	n	Number of milked cows	
*Tm*	min	Milk extraction (or flow) time	6–8 min
*Tri*	s	Early (initial) routine	Animal entry (*Ten*); bucket, trolley, or cluster positioning (*Tpos*), pre-dipping (*Tpre*), foremilk inspecting (*Tfor*), udder preparation (*Tup*), teat cups attach (*Tatt*)
*Trf*	s	Ending (final) routine	Teat cups removal (*Trem*), post-dipping (*Tpost*), bucket or tank empting (*Tempt*), relocation of the mobile parts of the milking system (*Trr*), animal exit (*Tex*)
*Tr*	s	Milking routine	30–180 s
*Td*	s	Downtime	Unproductive time due to unpredictable events (always present and split among *Tri, Trf* and *Trr*)
*Tc*	s	Milker’s time for milking each cow	*Tc* is always higher than, or equal to, the milking routine *Tr*
*Ng*	n	Number of stalls	Only in parlor
*Ng*	n	Number of milking groups	
*Tmw*	s	Milker waiting time	Unproductive routine time (for example due to the absence of the milking cluster removal

**Table 2 animals-10-00795-t002:** Routine of trolley and bucket systems.

Task		Time (s)		Routine Time (s)		Routine Time (min)	
	Input Data	Trolley	Bucket	Trolley	Bucket	Trolley	Bucket
Bucket or trolley positioning	Tpos	10	28	*Tri* = 67 + 8 *	*Tri* = 85 + 10 *	*Tri* = 1.25	*Tri* = 1.58
Pre-dipping	Tpre	15	15				
Foremilk inspection	Tfor	8	8				
Udder preparation	Tup	20	20				
Teat cups attach	Tatt	14	14				
Teat cups removal	Trem	25	25	*Trf* = 84 + 16 *	*Trf* = 84 + 10 *		
Post-dipping	Tpost	9	9			*Trf* = 1.67	*Trf* = 1.57
Bucket or container empting	Tempt	20	50				
Relocation of trolley	*Trr*	30	0				
Downtimes *	Td	24	20				

* Divided among *Tri*, *Trf* and *Trr* (this last only for trolley).

**Table 3 animals-10-00795-t003:** Routine of milk-lines.

		Time (s)		Routine Time (s)		Routine Time (min)	
Task	Input data	NOACR ^1^	YESACR ^2^	NOACR	YESACR	NOACR	YESACR
Milking group positioning	Tpos	23	15	*Tri* = 80 + 10 *	*Tri* = 72 + 6 *	*Tri* = 1.5	*Tri* = 1.3
Pre-dipping	Tpre	15	15				
Foremilk inspecting	Tfor	8	8				
Udder preparation	Tup	20	20				
Teat cups attach	Tatt	14	14				
Teat cups removal	Trem	25	-	*Trf* = 49 + 5 *	*Trf* = 21 + 6 *	*Trf* = 0.9	*Trf* = 0.45
Post-dipping	Tpost	9	9				
Group removal	Tempt	15	12				
Downtimes *	Td	15	12				

^1^ NOACR: without automatic cluster removal. ^2^ YESACR: with automatic cluster removal. * Divided between *Tri* and *Trf*.

**Table 4 animals-10-00795-t004:** Initial and final routine in trolley, bucket and milk-line systems.

Routine Time (min)	Trolley	Bucket	Milk-Line NOACR	Milk-Line YESACR
*Tri*	1.25	1.58	1.50	1.30
*Trf*	1.03	1.57	0.90	0.45

**Table 5 animals-10-00795-t005:** Routine of autotandem, herringbone and parallel parlors.

		Time (s)			Routine Time (s)		
Task	Input Data	Auto Tandem	Herring Bone	Parallel	Auto Tandem	Herring Bone	Parallel
Animal entry	*Ten*	15	8	6	*Tri* = 53 + 10 *	*Tri* = 40 + 6 *	*Tri* = 38 + 4 *
Pre-dipping	*Tpre*	15	9	9			
Foremilk inspecting	*Tfor*	6	6	6			
Udder preparation	*Tup*	7	7	7			
Teat cups attach	*Tatt*	10	10	10			
Post-dipping	*Tpost*	7	7	7	*Trf* = 9 + 4 *	*Trf* = 9 + 5 *	*Trf* = 9 + 4 *
Animal exit	*Tex*	2	2	2			
Downtimes *	*Td*	14 *	11 *	8 *			

* Divided between *Tri* and *Trf*.

**Table 6 animals-10-00795-t006:** Initial and final routine in autotandem, herringbone and parallel parlors.

Routine Time (min)	Auto Tandem	Herring Bone	Parallel
*Tri*	1.05	0.77	0.7
*Trf*	0.22	0.23	0.22

**Table 7 animals-10-00795-t007:** Descriptive statistic of the examined tie-stall dairy farms.

	Trolley		Bucket		Milk-Line NOACR		Milk-Line YESACR	
Dairy Farm Data	Average	SD	Average	SD	Average	SD	Average	SD
lactating cows (n)	24.7	2.52	34.3	6.0	35.0	8.89	50	-
milk flow time average (min)	6.4	0.12	6.7	0.17	6.9	0.10	7.1	-
milk flow time min (min)	5.5	0.12	5.4	0.30	5.8	0.17	6.8	-
milk flow time max (min)	7.1	0.12	7.2	0.06	7.6	0.06	7.6	-

**Table 8 animals-10-00795-t008:** Descriptive statistic of the examined dairy farms with a parlor.

	Autotandem		Herringbone		Parallel	
Dairy Farm Data	Mean	SD	Mean	SD	Mean	SD
lactating cows (n)	98.5	4.95	111.7	10.41	121.4	7.40
milk flow time average (min)	7.2	0.07	6.8	0.06	7.0	0.03
milk flow time min (min)	5.9	0.10	6.3	0.04	6.5	0.02
milk flow time max (min)	8.1	0.07	7.4	0.06	7.5	0.20

**Table 9 animals-10-00795-t009:** General linear model (GLM) statistic of *Tri* and *Trf* in the function of the milking system.

Origin		ss	df	Ms	F	*p*-Value
Correct model	*Tri*	7.169 ^a^	6	1.195	131.464	0.000
	*Trf*	22.204 ^b^	6	3.701	282.077	0.000
Intercept	*Tri*	64.386	1	64.386	7083.875	0.000
	*Trf*	28.773	1	28.773	2193.215	0.000
Milking system	*Tri*	7.169	6	1.1948	131.464	0.000
	*Trf*	22.204	6	3.701	282.076	0.000

ss: sum of squares; df: degrees of freedom; Ms: mean square; F: F-value. a: R-squared = 0.937 (R-squared corrected = 0.930). b: R-squared = 0.970 (R-squared corrected = 0.966).

**Table 10 animals-10-00795-t010:** Tukey post-hoc statistic of *Tri* and *Trf.*

Tri							Trf					
Milk_System	Test	Subset					Milk_System	Title	Subset			
	N	1	2	3	4	5		N	1	2	3	4
Parallel	15	0.70					Parallel	15	0.23			
Herringbone	9	0.75					Herringbone	9	0.25			
Autotandem	6		1.01				Autotandem	6	0.27	0.27		
Trolley	9			1.26			Milk-line YESACR	3		0.45		
Milk-line YESACR	3			1.30	1.30		Milk-line NOACR	9			0.94	
Milk-line NOACR	9				1.46	1.46	Bucket	9				1.57
Bucket	9					1.59	Trolley	9				1.68
Sign.		0.97	1.00	0.98	0.05	0.12	Sign.		0.99	0.07	1.00	0.61

**Table 11 animals-10-00795-t011:** Comparison of milking systems parameters in tie-stall.

Milking System	Tm = 6 min				Tm = 8 min			
	Tr	Tc	Ng	Nc	Tr	Tc	Ng	Nc
Trolley	2.91	8.91	2	13.46	2.91	10.91	2	5.50
Bucket	3.15	4.58	2	26.23	3.15	3.72	3	16.14
NOACR Milk-line	2.40	2.80	3	21.43	2.40	2.60	4	23.08
YESACR Milk-line	1.75	1.55	5	34.29	1.75	1.63	6	34.29

**Table 12 animals-10-00795-t012:** Comparison of milking systems parameters in different parlor types.

	*Tm* = 6 min					*Tm* = 8 min				
Parlor Type	*Tr*	*Tc*	*Ng*	*Ns*	*Nc*	*Tr*	*Tc*	*Ng*	*Ns*	*Nc*
Autotandem	1.27	1.27	6	(3 + 3)	47.4	1.27	1.27	8	(4 + 4)	47.4
Herringbone	1.00	1.00	12	(6 + 6)	60.0	1.00	1.00	16	(8 + 8)	60.0
Herringbone with swing-over	1.00	1.00	6	(6 + 6)	60.0	1.00	1.00	8	(8 + 8)	60.0
Parallel	0.92	0.92	14	(7 + 7)	65.5	0.92	0.92	18	(9 + 9)	65.5
Parallel with swing-over	0.92	0.92	7	(7 + 7)	65.5	0.92	0.92	9	(9 + 9)	65.5

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
