# Peer review of "Sizing Milking Groups in Small Cow Dairies of Mediterranean Countries"

_animals, 2020, doi:10.3390/ani10050795_

Round 1
Reviewer 1 Report
General comments: In this manuscript, the authors provide a description of the workflow in different milking/parlor systems. Overall, the manuscript is well structured and the authors thoroughly describe different scenarios of milking routines for the given parlor systems. This work is useful as it can serve as a reference for many regions in the European Union with similar farm sizes. Although the article is well written, it would benefit from language editing.
Specific comments:
I recommend to replace choice with choose throughout the manuscript each time you refer to the verb.
Please make sure that all abbreviations are introduced at first use. This is specifically true for the equations.
In several cases, the authors refers to “he” for either the farmer or the milker. I suggest to use either both gender terms or rephrase these sentences accordingly.
l 138: “must work on the breast” – do you mean the udder? – if so please rephrase.
l 148: “of our Country” – I suggest to name the country
l 176: “cows are handled in group” – suggest plural s for groups, same in line 257
Author Response
Dear Reviewer, we thank you for your support to improve our work. You may find our italic answers to your requests of revision. We want to really thank you for your efforts to refine on this paper.
- Overall, the manuscript is well structured and the authors thoroughly describe different scenarios of milking routines for the given parlor systems. This work is useful as it can serve as a reference for many regions in the European Union with similar farm sizes. Although the article is well written, it would benefit from language editing.
We provided a language revision of the manuscript.
Specific comments:
- I recommend to replace choice with choose throughout the manuscript each time you refer to the verb.
Done. It was a rough mistake, sorry.
- Please make sure that all abbreviations are introduced at first use. This is specifically true for the equations.
We really thank for this comment. There were some errors, indeed, due to the lack of a printer (we could not reach our workplace in this pandemic period). After printing the document, we could find all the abbreviation inconsistencies.
- In several cases, the authors refers to “he” for either the farmer or the milker. I suggest to use either both gender terms or rephrase these sentences accordingly.
To avoid gender bias (there are many female milkers also - and especially - in small dairy farms), the personal pronoun ‘he’ was corrected in ‘he/she’.
- l 138: “must work on the breast” – do you mean the udder? – if so please rephrase.
Yes, we mean udder. Corrected.
- l 148: “of our Country” – I suggest to name the country
Country was replaced by Italy.
- l 176: “cows are handled in group” – suggest plural s for groups, same in line 257
Done
Reviewer 2 Report
Brief summary
This paper investigated the modeling of milking routine for small dairy farm. Several equations were proposed to calculate the optimal sizing of milking unit according to the different milking system.
Broad comments
This paper investigated the modeling of milking routine for small dairy farm. The topic is rarely investigated in these years, but it lacks a clear comparison between the proposed models and the observed data in 20 small dairy farms here considered.
Specific comments
Title
Ok
Abstract
General comment. It summarizes the paper, but it does not report indications about the goodness of the formulas to march the on-field results.
- Introduction
Ok
- Material and methods
Line 110, Table 1: it is not clear to me what is Tm; is it a time or, as you wrote, a rate? If it is a rate it must be a quantity related to the time; otherwise, it is milking time (min)
Line 125: as above
Lines 147-149: if 2 milkings/d represents a standard situation for small herds, namely for mountain area, I am not sure that intervals between milkings are so regular; could you consider this aspect?
Lines 192-196: it lacks any statistical approach, starting from a descriptive statistic of the farms.
General comment. In my opinion, a paper like this does not represent the moment to repeat the classification of milking systems; therefore, I suggest to try to short this part using a possible reference. Additionally, it is necessary at least to know the sample and its descriptive statistics. Otherwise, it is difficult to consider this paper as an experimental study or, at least, a survey.
- Results
I do not understand the reason to report the time values both as second and as minutes within the Tables; where I fail?
General comment. According to my previous comment on Material and methods, this paper mainly describes the different milking systems and the formulas for their evaluation. I can’t find the presentation of results with their statistical parameters.
- Discussion
Line 307: probably, 4.2
General comment. Very few comments within each group of milking system (tiestall and parlor).
- Conclusions
Not exhaustive on the topic of the paper.
Tables
Table 1: it is not clear to me what is Tm; is it a time or, as you wrote, a rate? If it is a rate it must be a quantity related to the time; otherwise, it is milking time (min)
General comment. I do not understand the reason to report the time values both as second and as minutes; where I fail?
Figures
Good.
Figure 12: the reference [45] does not exist
References
References [23] and [24]: check the Authors’name.
References [24] and [38]: they seem the same thing.
Round 2
Reviewer 2 Report
Brief summary
This paper investigated the modeling of milking routine for small dairy farm. Several equations were proposed to calculate the optimal sizing of milking unit according to the different milking system.
Broad comments
This paper investigated the modeling of milking routine for small dairy farm. The topic is rarely investigated in these years, and now the paper offers a practically interesting tool for the farmers and their advisors.
Specific comments
Title
Ok
Abstract
Ok
Introduction
Ok
Material and methods
Ok
Results
Ok
General comment. The new tables offer an interesting tool for the farmer.
Discussion
General comment. Good.
Conclusions
Good.
Tables
Ok
Figures
Good.
References
Ok